# 2.5-dimensional covalent organic frameworks

Tomoki Kitano[1,2], Syunto Goto[1,2], Xiaohan Wang[1,2], Takayuki Kamihara[3], Yoshihisa Sei[3], Yukihito Kondo[4], Takumi Sannomiya[4], Hidehiro Uekusa[5] & Yoichi Murakami[1,2,6] ✉

Covalently bonded crystalline substances with micropores have broad applications. Covalent organic frameworks (COFs) are representative of such substances. They have so far been classified into two-dimensional (2D) and three-dimensional (3D) COFs. 2D-COFs have planar shapes useful for broad purposes, but obtaining good crystals of 2D-COFs with sizes larger than 10 μm is significantly challenging, whereas yielding 3D-COFs with high crystallinity and larger sizes is easier. Here, we show COFs with 2.5-dimensional (2.5D) skeletons, which are microscopically constructed with 3D bonds but have macroscopically 2D planar shapes. The 2.5D-COFs shown herein achieve large single-crystal sizes above 0.1 mm and ultrahigh-density primary amines regularly allocated on and pointing perpendicular to the covalently-bonded network plane. Owing to the latter nature, the COFs are promising as $CO_2$ adsorbents that can simultaneously achieve high $CO_2/N_2$ selectivity and low heat of adsorption, which are usually in a mutually exclusive relationship. 2.5D-COFs are expected to broaden the frontier and application of covalently bonded microporous crystalline systems.

Reticular chemistry[1] has significantly broadened the freedom of materials design. Following the development of metal-organic frameworks (MOFs)[2,3], covalent organic frameworks (COFs) emerged[4,5] as crystalline solids formed by stronger covalent bonds[6]. Although the directional covalent bonds limit the connection freedom much more strictly than coordination bonds in MOFs that have high nodal flexibility, the same limitation has enabled us to predict the structures of COFs relatively accurately[7–10]. So far, besides the development of new linkers and linkages[6–10], the structural diversity of COFs has been effectively widened by conformational transformations caused by the change in the host–guest interactions[11–13] and topological isomerisms of the skeletons[14,15]. However, to fully exploit the virtues of covalently reticulated systems that have higher stability and bond directionality, further fundamental expansion of the structural realm of COFs is strongly desired.

COFs have been classified into two general types. The first type is two-dimensional (2D) COFs typically constructed using planar two-handed (ditopic), three-handed (tritopic), and/or four-handed (tetratopic) building blocks, which are covalently bonded to each other to form quasi-atomic-thick layers that stack with each other by intermolecular forces[4,7–10]. Notably, some 2D-COFs had unreacted groups on the skeleton[16–22]; such groups can serve as reactive sites to which purposefully chosen chemical moieties can append to bring about new functionality. However, the size of the appending moieties is, in principle, limited to the pore size of the 2D-COFs because such unreacted groups point in an in-plane direction.

The second type is three-dimensional (3D) COFs constructed using stereoscopically extended bonds[5,23–25], which have often been fulfilled using tetrahedral–tetratopic building blocks such as tetrakis(4-aminophenyl)methane[23] (**TAM**, Fig. 1a). Owing to the third

[1]Laboratory for Zero-Carbon Energy, Institute of Integrated Research, Institute of Science Tokyo, Tokyo, Japan. [2]Department of Mechanical Engineering, Institute of Science Tokyo, Tokyo, Japan. [3]Facility Station Division, Open Facility Center, Institute of Science Tokyo, Yokohama, Japan. [4]Department of Materials Science & Engineering, Institute of Science Tokyo, Yokohama, Japan. [5]Department of Chemistry, Institute of Science Tokyo, Tokyo, Japan. [6]Department of Transdisciplinary Science & Engineering, Institute of Science Tokyo, Tokyo, Japan. ✉e-mail: murakami.y.af@m.titech.ac.jp

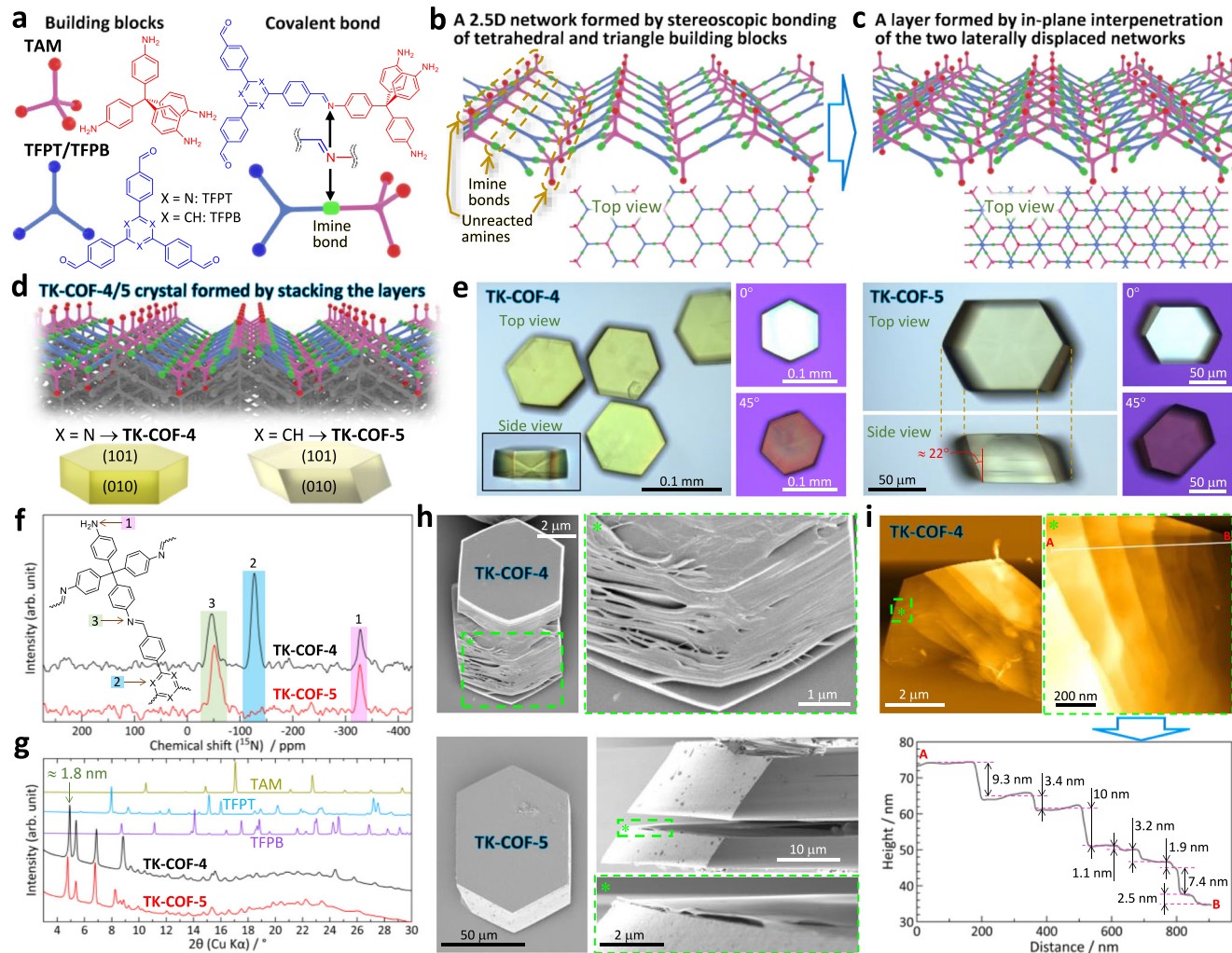

**Fig. 1 | Formation and fundamental characterizations of the COFs developed.**
**a** Building block molecules used and an imine bond formed between them.
**b** Schematic of the laterally extended network stereoscopically constructed by imine bonds between **TAM** and **TFPT/TFPB**. Here, one-fourth of the amines of **TAM** are unreacted and point to the out-of-plane direction. **c** Schematic of a layer formed by an in-plane interpenetration of the two laterally displaced networks shown in (**b**). **d** Schematic of crystals formed by stacking the layers shown in (**c**). **e** Optical micrographs of **TK-COF-4** and -**5** crystals in *o*-dichlorobenzene. The polarized optical images were obtained with a crossed-Nicols configuration. **f** Solid-state $^{15}$N CP/MAS NMR spectra. **g** PXRD patterns obtained from **TK-COF-4** and -**5** crystals in acetonitrile, compared with patterns from the powders of **TAM, TFPT**, and **TFPB**. **h, i** SEM and AFM images, respectively; the crystals of **TK-COF-4** were mechanically treated before the observations (see Methods). A cleaved crystal of **TK-COF-5** was viewed from the side in panel **h**.

dimension available for bond extensions, 3D-COFs have richer structural diversity[23–25]. A unique class is 1D ribbons reported by Yaghi and co-workers (COF-76) with unreacted amines on the edges of the ribbons, which were subsequently converted to a 2D-COF[26].

Considering the crystallization problem[6,9,23,24,27] that has persistently afflicted COFs, one advantage of 3D-COFs over 2D-COFs is the relative ease of obtaining single crystals. The majority of 2D-COFs were highly polycrystalline and their crystal shapes were not well recognizable using a scanning electron microscope (SEM), except for a few cases[13,28–30]. In contrast, single-crystal 3D-COFs with sizes exceeding 10 μm have recently been reported[15,30–34], albeit 3D-COF crystals over 100 μm are still rare[30,32,34]. Regarding the influence of dimensionality on the crystallinity, Haase and Lotsch[35] insightfully remarked that the covalent connectivity in three independent directions in 3D-COFs is considered to lead to improved crystallinity as compared to 2D-COFs.

So far, most COF research has been devoted to 2D-COFs, the partial reason for which may be their planar shape, which is useful for broad applications including separation membranes[36] and nanolayer devices[37,38]. However, as mentioned above, 2D-COFs have been severely challenged by the crystallization problem that has hampered the formation of large single crystals. So far, the record single-crystal size of 2D-COFs attained with one-step growth has been limited to *ca.* 3–10 μm[29,30]; to attain further larger sizes, a recent X-ray structural study used two-step growth in which seed crystals were re-grown in a fresh solution, yielding pyrene 2D-COF crystals with sizes of 15–45 μm[13].

Herein, to show one effective approach to settle the dilemma of balancing the advantages and disadvantages of 2D *vs.* 3D COFs, we report a new class of COFs that can achieve the single-crystal size of 0.1 mm while having practically advantageous layered structures. The discovered COFs are unprecedentedly constructed by stereoscopic (*i.e.,* microscopically 3D) bonds that extend laterally to result in 2D reticulated skeletons (*i.e.,* macroscopically 2D). Through this discovery, we present a structural concept of 2.5D-COFs based on the fact that such a skeleton type does not match the normal depictions[7,9,10,24,25] that have been used to describe 2D and 3D COFs. Specifically, this structural concept provides a new avenue to simultaneously realize advantages of 2D and 3D COFs, typical of which are a multitude of applications and ease of growing large single crystals, respectively, by reconciling the previously established binary distinction between 2D and 3D concepts.

We synthesized such COFs by bonding tetrahedral–tetratopic building blocks and triangle–tritopic building blocks (Fig. 1a, b), the combination of which has generated exclusively 3D-COFs previously[5,24,25,39,40]. The virtue of the 2.5D concept lies in the ability to generate large single crystals owing to the essentially 3D nature (remember the Haase and Lotsch's remark[35] above) and hence the effectiveness to address the crystallinity problem that has particularly afflicted 2D-COFs.

Notably, the new class of COFs are the first layered COFs that (i) comprise tetrahedral building blocks, (ii) reach single-crystal sizes of 0.1 mm, (iii) have high-density unreacted functionalities that point perpendicular to the planar network, and (iv) exhibit in-plane inter-penetration formed by laterally displaced networks. Among them, (iii), high-density primary amine groups pointing out-of-plane, is significant because purposefully chosen chemical moieties could be appended perpendicularly to the covalently reticulated crystalline layers, different from the aforementioned 2D-COFs with unreacted groups in which the size of the appendable moieties is, in principle, limited by the pore size of the 2D-COFs.

## Results and discussion

We combined **TAM** as a tetrahedral–tetratopic building block and 4,4′,4′′-(1,3,5-triazine-2,4,6-triyl)tris[benzaldehyde] (**TFPT**) or 1,3,5-tris(4-formylphenyl)benzene (**TFPB**) as triangle–tritopic building blocks (Fig. 1a, left). After the formation of imine bonds (Fig. 1a, right) in solution at 22 °C (see Methods), a stereoscopic but laterally extended network was formed with one-fourth of the amines of **TAM** unreacted (Fig. 1b). Owing to the tetrahedral geometry, all unreacted amines point normal to the extended network. A lateral interpenetration of the two networks, which is reported for the first time herein, forms one layer (Fig. 1c). The stacking of the layers forms **TK-COF-4** and **TK-COF-5** for **TFPT** and **TFPB**, respectively (Fig. 1d).

Although 2 days is sufficient to grow highly crystalline COFs, growth for 6–11 days at 22 °C yielded translucent crystals of sizes up to *ca.* 0.1 mm (Fig. 1e and Supplementary Table 1). Their status as single crystals was indicated by polarized microscopy (Fig. 1e). The Fourier transform infrared (FT-IR) spectra (Supplementary Fig. 1) indicated the disappearance of aldehyde C = O (**TFPT**: 1710 cm⁻¹, **TFPB**: 1687 cm⁻¹) and emergence of imine C = N (**TK-COF-4**: 1622 cm⁻¹, **TK-COF-5**: 1623 cm⁻¹) signals, evidencing a full consumption of the aldehyde groups. The existence of primary amines in the COFs was evidenced by the solid-state ¹⁵N cross-polarization magic-angle-spinning (CP/MAS) NMR spectra (Fig. 1f), X-ray photoelectron spectroscopy (XPS; Supplementary Fig. 3), and more directly by the single-crystal X-ray diffraction (SCXRD) results shown below. The powder X-ray diffraction (PXRD) patterns acquired in acetonitrile exhibited low-angle peaks at $2\theta = 4.8$–$4.9°$ (Fig. 1g) corresponding to the period of *ca.* 1.8 nm, whereas those acquired in the dried state showed peaks at $2\theta \approx 4.6$–$4.8°$ (Supplementary Fig. 4) corresponding to the period of *ca.* 1.9 nm.

The layered nature of **TK-COF-4** and **-5** was revealed by scanning electron microscope (SEM) images (Fig. 1h), in which **TK-COF-4** crystals were mechanically treated before the observation (see Methods for the treatment). The AFM images from the mechanically treated crystal indicate nanometer-level plateaus (Fig. 1i and Supplementary Fig. 6 for **TK-COF-5**).

The large crystal size afforded direct determinations of their structures by SCXRD. From the analysis, the unit cell parameters were found for **TK-COF-4** ($P2_1/n$ space group, $a = 9.8024(4)$ Å, $b = 36.6977(9)$ Å, $c = 18.2022(5)$ Å, $\alpha = 90°$, $\beta = 99.608(3)°$, $\gamma = 90°$, and $V = 6455.95$ Å³) and **TK-COF-5** ($P2_1/n$ space group, $a = 10.2481(7)$ Å, $b = 37.3978(18)$ Å, $c = 18.6451(8)$ Å, $\alpha = 90°$, $\beta = 103.078(5)°$, $\gamma = 90°$, and $V = 6960.51$ Å³).

The determined crystal structures (Fig. 2a) revealed two new features as COFs, which are in-plane interpenetration of two laterally displaced skeletons (topology: **hcb**) that form a layer and regularly positioned free primary amines pointing in an out-of-plane direction to

the layer (Fig. 2a; *cf.* Figure 1c). For the latter, the high area density of the amines (4.9 and $4.6 \times 10^5$ μm⁻² layer⁻¹ for **TK-COF-4** and **-5**, respectively) is noted. This area density is much higher compared to previous 2D-COFs in which primary amines were left unreacted, whereas it is comparable to or lower than that of 2D-COFs to which primary amines were appended by post-synthesis modification (Supplementary Table 6). These features stem from the novel skeleton type formed by stereo-scopically and non-stoichiometrically bonding tetrahedral and triangular building blocks. The results of elemental analyses (Supplementary Table 2), ¹³C CP/MAS NMR (Supplementary Fig. 2), and ¹⁵N CP/MAS NMR (Fig. 1f) are consistent with the crystal structures determined by the SCXRD. In addition, the simulated PXRD patterns generated from the crystal structures are consistent with the measured PXRD patterns (Supplementary Figs. 8 and 10), from which the peaks at 4.8–4.9° (Fig. 1g) are assigned to the (020) planes. The determined crystal structures show hydrogen bonds between H of the primary amine and N of the imine in the adjacent layer (Fig. 2a, right bottom), the H···N and inter-N distances of which ($\cong 2.4$–$2.7$ and $\cong 3.2$–$3.3$ Å, respectively) were similar to those reported for N–H···N bonds[26,41,42].

Furthermore, we observed the crystals with high-resolution transmission electron microscopy (HR-TEM) in vacuum. The images showed stripes with a period of *ca.* 1.9 nm in the [010] direction (Fig. 2b, c), which agreed well with the period of *ca.* 1.9 nm indicated by the peaks found at 4.6–4.8° of the PXRD patterns acquired in the dried state (Supplementary Fig. 4); the solvent removal caused slight deformation of the framework (Supplementary Figs. 15 and 17).

Notably, to the best of our knowledge, **TK-COF-4** and **-5** achieved the largest single-crystal sizes (≥0.1 mm, Fig. 1e) for layered COFs, far larger than the previous record size of 3–10 μm attained by one-step growth[29,30] and 15–45 μm by two-step recrystallization[13]. We surmise that one of the reasons for the far larger sizes achieved by **TK-COF-4** and **-5** could lie in the high symmetry of **TAM**. That is, the consumption of three amine hands out of four hands renders four degrees of multiplicity in the choice of hands, and the threefold axial symmetry of **TAM** further renders three degrees of rotational multiplicity, resulting in many equivalent postures of **TAM** to be incorporated into the stereoscopic skeletons constituting these COFs.

Hereafter, we show several fundamental practical properties of **TK-COF-4** and **-5**. First, thermogravimetric analyses (TGA) showed that these COFs have outstanding thermal stability in the air and N₂ (Fig. 3a). The unchanged PXRD patterns and sample colors, after being heated using the same TGA apparatus and heating conditions, evidenced that these COFs are structurally stable at least up to 300 °C in N₂ and 200 °C in the air (Fig. 3b and Supplementary Figs. 18 and 19); the color turned brownish and the PXRD intensity decreased after being heated to 300 °C in the air. Such thermal stabilities are much higher than those of previous 2D-COF with unreacted primary amines, **NH₂-Th-Tz COF**[20]. We attribute the observed high thermal stability to the high crystallinity that may have protected the amines and imines by the regular and tight hydrogen bonds (see Fig. 2a).

The N₂ adsorption-desorption isotherms at 77 K exhibited type-I behaviors (Fig. 3c). The analyses revealed the BET surface areas of *ca.* 656 and 416 m² g⁻¹ for **TK-COF-4** and **-5**, respectively (Supplementary Fig. 22) and distinct pore size at around 0.6–0.7 nm (Supplementary Fig. 23), reflecting their microporous nature. The hysteresis in the isotherms are similar to those found for previous flexible and crystalline COFs[11–13,30], and some of the reports[11,12] suggested the framework flexibility as the potential reason for the phenomenon. Because the present **TK-COF** can deform by solvent removal (Supplementary Fig. 17), we surmise that the flexibility was a possible reason for the hysteresis.

To exemplify a potential application, we evaluated these COFs for CO₂ capture. As expected, **TK-COF-4**, which has an additional N on the hydrazine ring, exhibited a higher CO₂ uptake at 100 kPa (55 and

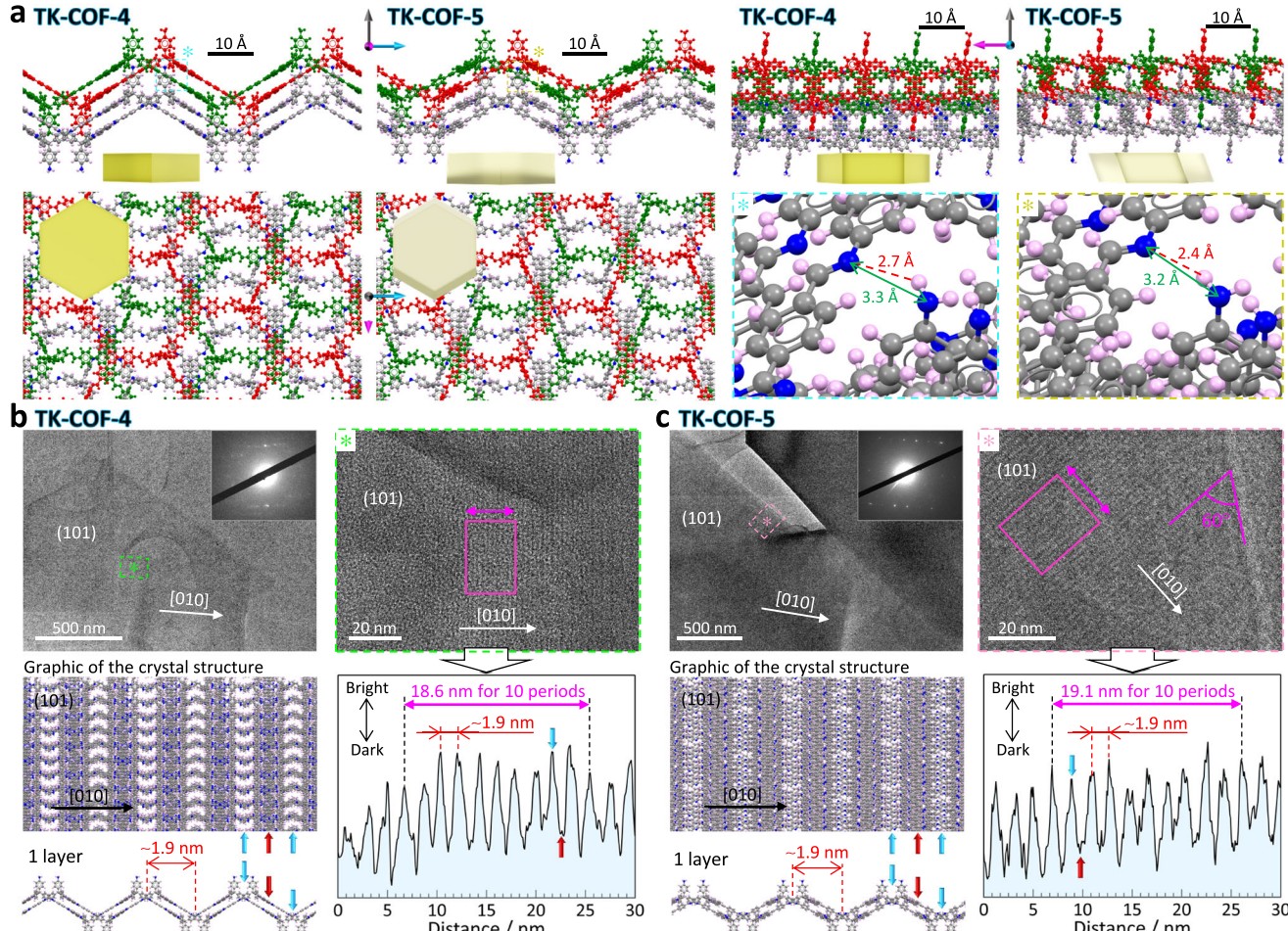

**Fig. 2 | Structures of TK-COF-4 and -5. a** Crystal structures determined by SCXRD measurements. Red and green nets show two covalently reticulated nets that laterally interpenetrate each other to form one layer. The layers below them are illustrated using gray for C, light pink for H, and blue for N. Each panel is accompanied by orthogonal arrows to indicate the direction of the view, along with the corresponding view of the crystal. The right-bottom panels magnify the hydrogen bonds between the primary amine and its adjacent imine. **b, c** HR-TEM images and the comparisons with the structural models determined by the SCXRD measurements. Light blue (red) arrows indicate the bright (dark) positions in the images.

30 cm³ g⁻¹ STP at 273 and 298 K, respectively; Fig. 3d) than **TK-COF-5**; these uptake values are similar to those of previous COFs for which $CO_2$ adsorption characteristics were reported[43] (Supplementary Table 11 and references therein). Theoretically, 100% capture of $CO_2$ by the primary amines in **TK-COF-4** and **-5** corresponds to the $CO_2$ uptake of 33.2 and 34.0 cm³ g⁻¹ STP, respectively. These COFs exhibited high adsorption selectivity to $CO_2$ over $N_2$ (Fig. 3d). The $CO_2/N_2$ selectivity was calculated at $CO_2:N_2 = 15:85$ (mol) using ideal adsorbed solution theory (IAST)[44,45] and found to be *ca.* 100 or higher (Fig. 3e).

One central problem of the current aqueous amine method is the excessively large heat of adsorption ($Q_{st}$) of $CO_2$ to the solute amines of around 80–120 kJ/mol·$CO_2$[45–47], which has necessitated large thermal energy input during the regeneration process to strip $CO_2$ from the amines. Because such a large $Q_{st}$ is unnecessary when industrial flue gases that typically comprise 5–50%[48] $CO_2$ are targeted, the smaller $Q_{st}$ reduces the energy cost that is required to run the adsorption-regeneration cycles[45,49,50] as long as the selectivity to $CO_2$ is sufficiently high. Meanwhile, the limitations in the material's adsorption capacity of $CO_2$ could be coped with by increasing the amount of adsorbent loaded, because the impact of the adsorbent's $CO_2$ uptake capacity (mol·$CO_2$ per kg) on the capture cost (price per ton·$CO_2$) is weak in a system scale (see Figure 8 of ref. 49 and Figure 7 of ref. 51).

Therefore, particularly considering the energy cost, we assume that COFs with lower $Q_{st}$ and higher $CO_2/N_2$ selectivity are preferable. Notably, achieving low $Q_{st}$ and high $CO_2/N_2$ selectivity is usually mutually exclusive[49,50] because a higher selectivity to $CO_2$ is often caused by a stronger grasp of $CO_2$ molecules, as indicated by a plot previously compiled for MOFs (*cf.* Figure 2d of ref. 52) and the plots we have compiled for MOFs (Supplementary Fig. 35) and COFs (Fig. 3g below).

In this study, the $Q_{st}$ values (see Methods for the calculation) were nearly constant over 0–1 bar and their averages ($Q_{st(av)}$) were 25.5 and 25.3 kJ mol⁻¹ for **TK-COF-4** and **-5**, respectively (Fig. 3f). Such low $Q_{st(av)}$ values are considered to be caused by the low basicity of aromatic amines, from which we attribute the present capture mechanism to physisorption[53,54]. We plotted $Q_{st(av)}$ and the IAST selectivity values at 1 bar (Fig. 3e) and compared them with the results of previous COF literature that reported $Q_{st}$ and IAST selectivity at 273 or 298 K for $CO_2:N_2 = 15:85$, which is the most popular ratio used previously[43,50] (Fig. 3g and Supplementary Table 11 and references therein). With the aforementioned mutually-exclusive tendency seen, Fig. 3g shows that **TK-COF-4** and **-5** have outstanding characteristics compared with the previous COFs because of the simultaneous achievements of low $Q_{st}$ and high $CO_2/N_2$ selectivity. The advantage is also seen when

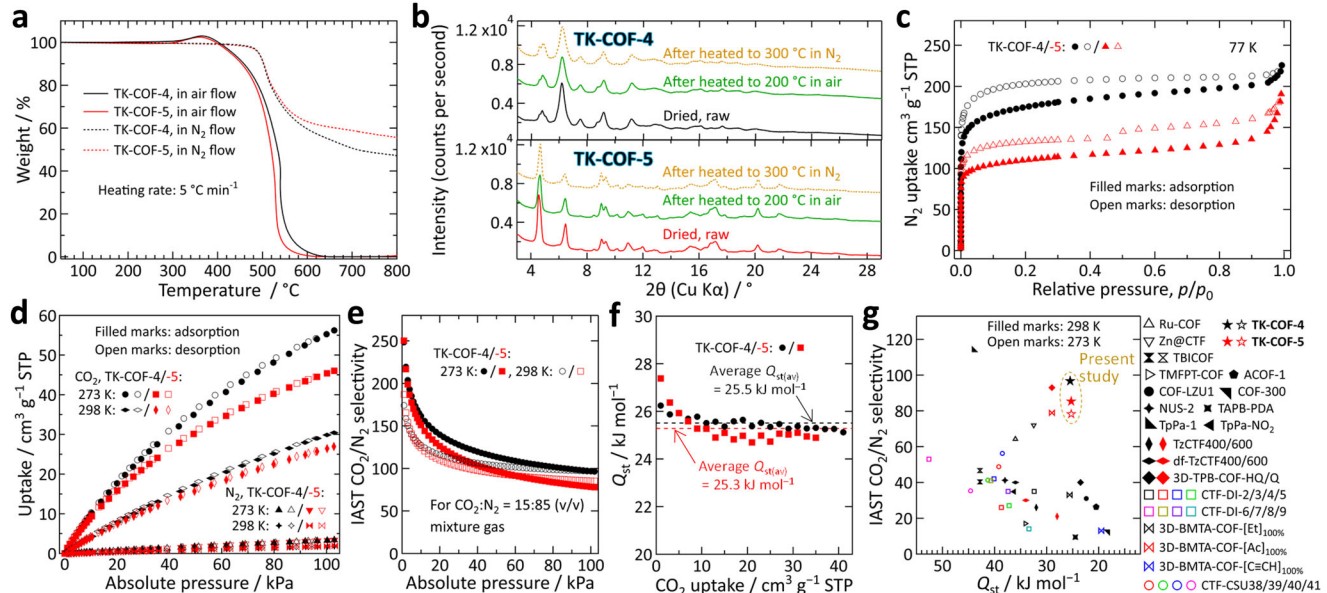

**Fig. 3 | Fundamental practical properties of TK-COF-4 and -5. a** TGA curves acquired in air and $N_2$ at a heating rate of 5 °C min⁻¹. **b** PXRD patterns from the samples after heating to 200 and 300 °C in the air and $N_2$, respectively, using the same TGA apparatus and heating rate. **c** $N_2$ adsorption-desorption isotherms at 77 K. **d** $N_2$ and $CO_2$ adsorption-desorption isotherms at 273 and 298 K. **e** IAST $CO_2$ selectivity calculated for a mixed gas of $CO_2:N_2 = 15:85$ at 273 and 298 K; see

Methods. **f** Isosteric heat of adsorption of $CO_2$ determined from the temperature dependence of the adsorption isotherms at 273, 285.5, and 298 K; see Methods. **g** Comparison of the relationships of the IAST $CO_2/N_2$ selectivity at a 15:85 ratio ($S_{CN(15:85)}$) at 100 kPa *vs.* $Q_{st(av)}$ for **TK-COF-4/-5** and those of $S_{CN(15:85)}$ *vs.* $Q_{st}$ for previous COFs reported with $CO_2$ adsorption characteristics at 100 kPa or 1 bar.

compared with previously reported MOFs (Supplementary Section 3.2) and porous organic polymers (Supplementary Section 3.3). This feature may have arisen from the unique high-density primary amines arranged on these 2.5D skeletons. Notably, the ordered crystalline structure of the present COF has an advantage in the faster $CO_2$ adsorption kinetics over a reference material with lower crystallinity (Supplementary Section 2.15). Furthermore, thermal treatment in air at 100 °C for 12 h did not affect the $CO_2$ adsorption capacity (Supplementary Fig. 20) or the PXRD pattern and FT-IR spectrum (Supplementary Fig. 21), indicating high stability against oxidative degradation, unlike previous silica-supported amine sorbents that showed a significant decrease of $CO_2$ adsorption capacity as a result of such thermal treatment[55]. Considering the high thermal stability even in the presence of oxygen, the present COFs could be useful for the stated purpose.

Finally, we comment on the factors that may contribute to the formation of the 2.5D structure. Previous COFs made by combining tetrahedral and triangular building blocks were 3D-COFs with either **bor**[5,40] or **ctn**[5,39] topology, the reported densities of which were low (0.17–0.41 g cm⁻³ in ref. 5; 0.43–0.53 g cm⁻³ in ref. 39; 0.13 g cm⁻³ in ref. 40). In contrast, the density of **TK-COF-4/-5** was much higher (*ca.* 0.7–1 g cm⁻³, Supplementary Table 5), implying higher thermodynamic stability. From our structural energy calculations using the COMPASS III force field, the total energies of **TK-COF-4/-5** were much lower than those of the hypothetical **bor** and **ctn** COFs constructed using **TAM** and **TFPT/TFPB** because the non-bonding energies of **TK-COF-4/-5** were much lower than those of these hypothetical 3D-COFs (Supplementary Section 2.16). Therefore, we hypothesize that the use of building blocks that would cause large inter-layer van der Waals attractions (such as π···π) and hydrogen bonds (such as N–H···N) shall increase the possibility of forming the 2.5D structure.

In summary, this article reported a new class of COFs with 2.5D−microscopically 3D but macroscopically 2D−skeletons, embodied by **TK-COF-4** and **-5**. These COFs have unique features owing to their constitution of tetrahedral−tetratopic building blocks and their non-stoichiometric bonding with trigonal−tritopic building blocks,

whereas all previous reports that combined such building blocks have generated 3D-COFs. Although we posed the hypothesis above, the range of the generality of this approach or requisites for constructing such 2.5D structures should be clarified in the future as an open question. Because of the structural uniqueness, unreacted free primary amines are, for the first time, aligned perpendicular to the laterally extended networks reticulated by covalent bonds. This feature has resolved the limitation of 2D-COFs in which the size of the chemical moieties appended to the unreacted functionality is, in principle, limited by the size of the pore. Owing to the 3D bonds comprising this class of COFs, the record-large single-crystal sizes of up to 0.1 mm were achieved for layered COFs, which is expected to be advantageous for broad applications. Thus, 2.5D-COFs are considered to have resolved the limitations of 3D-COFs (a difficulty in providing a planar shape that is useful for applications) and 2D-COFs (a difficulty in achieving large single crystals). High thermal stability and $CO_2$ capture performances, as well as the high-density primary amines arranged normally on the planar skeletons, may render these COFs useful for broad applications. Such a new class of COFs will open a new domain of covalently-reticulated crystalline systems, further extending the growing frontier of reticular chemistry.

## Methods
### COF synthesis
**Chemicals used.** We purchased tetrakis(4-aminophenyl)methane (**TAM**, ≥95%) from Accela ChemBio (USA) as a tetrahedral−tetratopic building block and 4,4′,4″-(1,3,5-triazine-2,4,6-triyl)tribenzaldehyde (**TFPT**, ≥97%) and 1,3,5-tris(4-formylphenyl)benzene (**TFPB**, 98%) from Jilin Province Extension Technology (China) as triangle−tritopic building blocks. They were used without further purification.

For the growth of COFs, we used 1,4-dioxane (99.5+%, FUJIFILM Wako Chemicals, Japan) and *o*-dichlorobenzene (*o*-DCB; >99.0%, TCI, Japan) as the solvents, acetic acid (AcOH; 99.5+%, FUJIFILM Wako Chemicals) as the catalyst, and either aniline (≥99.5%, Sigma-Aldrich, USA) or *m*-toluidine (99.0+%, FUJIFILM Wako Chemicals) as the modulator.

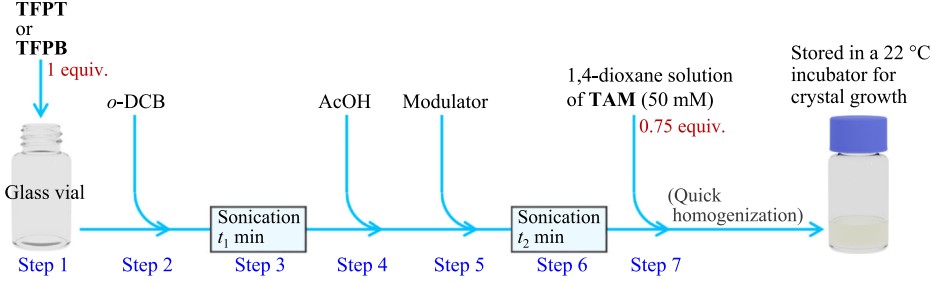

**Fig. 4 | Procedure for the sample preparation.** The preparation procedure consists of the following seven steps. Step 1: weighing of **TFPT** or **TFPB** into a glass vial. Step 2: addition of *o*-dichlorobenzene (*o*-DCB) into the vial. Step 3: sonication to disperse **TFPT** or **TFPB**. Step 4: addition of acetic acid (AcOH). Step 5: addition of modulator (aniline or *m*-toluidine). Step 6: sonication to homogenize the mixture. Step 7: addition of 1,4-dioxane solution of **TAM** (50 mM) into the vial. After the mixture was quickly homogenized, the vial was capped and stored in an incubator at 22 °C. See Methods and Table 1 for further details.

## Table 1 | Summary of sample preparation conditions

| Preparation condition | TFPT or TFPB 1 equiv. | Amount of TFPT/ TFPB taken | Volume of *o*-DCB taken | Time for sonication ($t_1$) | Amount of AcOH added | Modulator type, amount added | Time for sonication ($t_2$) | Volume of TAM solution (50 mM in 1,4-dioxane) added 0.75 equiv. |
|---|---|---|---|---|---|---|---|---|
| | Step 1 | | Step 2 | Step 3 | Step 4 | Step 5 | Step 6 | Step 7 |
| I | TFPT | 0.024 mmol 9.44 mg | 2.16 mL | 10 min | 182 equiv. 252 µL | Aniline 62 equiv. 136 µL | 3 min | 360 µL |
| II | TFPT | 0.048 mmol 18.88 mg | 4.32 mL | 10 min | 60.7 equiv. 168 µL | Aniline 29 equiv. 127 µL | 3 min | 720 µL |
| III | TFPB | 0.024 mmol 9.36 mg | 2.16 mL | 5 min | 182 equiv. 252 µL | *m*-Toluidine 42 equiv. 110 µL | 3 min | 360 µL |
| IV | TFPB | 0.048 mmol 18.74 mg | 4.32 mL | 5 min | 182 equiv. 504 µL | *m*-Toluidine 20 equiv. 105 µL | 3 min | 720 µL |

**Preparation procedure of the COFs.** Synthesis of the COFs was carried out in a step-by-step manner, which consisted of seven steps (Steps 1–7) described in Fig. 4. The specific conditions for each step are summarized in Table 1. As shown in the table, we used two conditions for each COF type (Conditions I and II for **TK-COF-4** and Conditions III and IV for **TK-COF-5**), depending on the purpose of the measurement. From our experience, Conditions I and III generated larger crystals with lower yield, which was suitable for optical microscope observations and single-crystal X-ray diffractions, whereas Conditions II and IV generated smaller crystals with higher yield, which was suitable for characterizations that needed a sizable sample amount. We selected conditions depending on the purpose of the measurement (see Supplementary Table 1). We have confirmed that Conditions I and II, as well as Conditions III and IV, generate the same product, as supported by the identities of the PXRD patterns (see Supplementary Fig. 5).

Specifically, the COF synthesis was conducted following the steps shown in Fig. 4 and conditions summarized in Table 1. We define the mole of **TFPT** or **TFPB** as "1 equiv." First, the powder of **TFPT** or **TFPB** was taken in a screw-cap glass vial (capacity: 5 or 10 mL). Then, *o*-DCB was added to the vial, which was sonicated using an ultrasonic bath (3510-DTH, Branson). Next, AcOH and modulator (either aniline or *m*-toluidine) were added to the vial, after which an additional sonication was applied. Subsequently, a 1,4-dioxane solution of **TAM** at 50-mM concentration, which was prepared in a separate vial and passed through a PTFE filter (pore size: 200 nm, SLLGX13NL, Merck-LG), was added to it. Finally, the mixture was homogenized quickly either by sonication for *ca.* 1 min or by gently shaking the vial by hand. The capped vial was stored in a temperature-controlled incubator at 22 °C until crystals with sufficient size for intended measurement were generated (see Supplementary Table 1).

## Sample characterizations
**Washing treatment of crystals before characterizations.** First, we replaced the growth solution in the vial with a fresh *o*-DCB twice to remove acetic acid, modulator, and intermediates; most COF crystals were adhered on the bottom or inner wall of the vial. Subsequently, the *o*-DCB was replaced twice by a fresh *N,N*-dimethylacetamide (DMA; >99.0%, TCI), after which the COF crystals adhered on the inner vial wall could easily be detached with gentle mechanical scrubbing of the inner wall using the tip of a Pasteur pipette. Then, the DMA that suspended the crystals was divided among several Eppendorf tubes (capacity: 1.5 mL) and underwent centrifugation. For sample characterizations that required a bulk amount of dried crystals, we replaced the supernatant DMA twice by a fresh toluene (99.5+%, FUJIFILM Wako Chemicals) to ensure the removal of DMA. Our use of toluene here was because of a previous report[56] that the use of a nonpolar solvent could minimize the surface-tension-induced damage that might be caused to COF pores during the subsequent drying process. Finally, the crystals were collected on a filter paper and dried in a vacuum at 80 °C for 12 h with a flow of 50 sccm of dry nitrogen.

**Optical microscopy.** Optical micrographs (Fig. 1e) were obtained using a polarized microscope (BX53, Olympus) equipped with a CMOS camera. Polarized microscope images were acquired using a rotatable sample stage and a pair of polarizers that were adjusted in the crossed-Nicols configuration with a retardation plate of 530 nm.

**Fourier-transform infrared (FT-IR) spectroscopy.** Fourier-transform infrared (FT-IR) spectra were acquired using FT/IR-6100 (JAFCO) with a single-reflection attenuated total reflectance unit (prism material: germanium). During the measurements, the inside of the sample

chamber was evacuated under vacuum to exclude the effect of ambient air. Before the measurement, COFs were dried in a vacuum at 80 °C for 12 h with a flow of 50 sccm of dry nitrogen.

**Solid-state $^{13}C$ and $^{15}N$ nuclear magnetic resonance (ss-NMR) spectroscopy.** Solid-state nuclear magnetic resonance (ss-NMR) measurements were conducted with magic-angle spinning (MAS) on an FT-NMR spectrometer (JNM-ECA400, JEOL). The spectra were obtained using a 9.39 T standard-bore magnet with Larmor frequencies of 399.78, 100.53, and 40.51 MHz for $^{1}H$, $^{13}C$, and $^{15}N$ nuclei, respectively. Both $^{13}C$ and $^{15}N$ cross-polarization (CP) MAS experiments were carried out on a standard 3.2-mm double-resonance HX probe with sample spinning rates of 18 kHz for $^{13}C$ and 14.5 kHz for $^{15}N$. CP/MAS experiments were carried out using RAMP-CP[57] and $^{1}H$ TPPM[58] decoupling with $^{1}H$ RF amplitudes of *ca.* 100 kHz for decoupling and *ca.* 88 kHz for CP, a contact time of 3 ms, and a pulse delay of 1 s. $^{15}N$ CP/MAS experiments were acquired with RF amplitudes of *ca.* 100 kHz for $^{1}H$ TPPM decoupling, *ca.* 40 kHz ($^{1}H$) and *ca.* 11–20 kHz ($^{15}N$) for RAMP-CP, a contact time of 3 ms, and a pulse delay of 1 s. The $^{13}C$ and $^{15}N$ chemical shifts were referenced relative to tetramethylsilane and $CH_3NO_2$, respectively, at 0 ppm. Before the measurement, COFs were dried in a vacuum at 80 °C for 12 h with a flow of 50 sccm of dry nitrogen.

**X-ray photoelectron spectroscopy (XPS).** X-ray photoelectron spectroscopy (XPS) on the samples were conducted using a photoelectron spectrometer (VersaProbe III, ULVAC-PHI) with Al Kα irradiation at the Open Facility Center in Tokyo Institute of Technology (Institute of Science Tokyo after October 2024). Because COFs are electric insulators, charge compensation was carried out during measurements. We conducted curve fitting to the obtained spectra (Supplementary Fig. 3) using PHI MultiPak® software. Before the measurement, COFs were dried in a vacuum at 80 °C for 12 h with a flow of 50 sccm of dry nitrogen.

**Powder X-ray diffraction (PXRD) measurements.** PXRD measurements were conducted using an automated X-ray diffractometer (SmartLab, Rigaku) with Cu Kα radiation ($\lambda = 1.54184$ Å) at 40 kV and 50 mA. The PXRD patterns of the building block molecules (**TAM, TFPT,** and **TFPB**) were acquired for the powder packed into a borosilicate glass capillary (diameter: 0.7 mm) at a scan rate and rotation speed of 1.0° min$^{-1}$ and 120 rpm, respectively; the average of three scans was taken to increase the signal-to-noise ratio. The PXRD patterns of COFs in acetonitrile, sealed in a borosilicate glass capillary (diameter: 0.5 mm), were acquired at a scan rate and rotation speed of 0.2° min$^{-1}$ and 120 rpm, respectively; the average of three scans was taken. The PXRD patterns of COFs in a dried state were acquired in reflection mode in Bragg–Brentano geometry at a scan rate of 0.2° min$^{-1}$; the average of two scans was taken.

**Scanning electron microscopy (SEM).** For sample observation by scanning electron microscopy (SEM), we used SU8000 Type II (Hitachi High-Tech) in the Open Facility Center at Tokyo Institute of Technology (Institute of Science Tokyo after October 2024) with an acceleration voltage of 1.0 kV and a working distance of 8 mm. The mechanical treatment on the **TK-COF-4** crystals used in Fig. 1h, i and **TK-COF-5** crystals used in Supplementary Fig. 6 (for AFM, see below) were conducted as follows. First, the crystals were washed with *o*-DCB and DMA according to the procedure described above. Next, the crystals were transferred to propylene carbonate held in a screw-capped glass vial (capacity: 6 mL) and then mechanically agitated by a PTFE-coated magnetic stirring rod on a magnetic stirrer for 2 days at room temperature. Subsequently, the vial was sonicated using an ultrasonic bath (3510-DTH, Branson) for 30 min. Immediately after this, the treated crystals dispersed in propylene carbonate were cast onto a piece of

silicon wafer (*ca.* 1 cm × 1 cm) and then dried in a vacuum at 80 °C for about 18–24 h with a flow of 50 sccm of dry nitrogen. To prepare the **TK-COF-5** crystals observed in Fig. 1h, the crystals dispersed in propylene carbonate without mechanical agitation were cast onto a piece of silicon wafer (*ca.* 1 cm × 1 cm) and then dried in a vacuum at 80 °C for about 18–24 h with a flow of 50 sccm of dry nitrogen.

**Atomic force microscopy (AFM).** We used Cypher S (Oxford Instruments) using sample crystals mechanically treated according to the method described above. We used AC200-TS (Olympus) for AFM probes. The image processing and analysis were conducted using Gwyddion® software.

**Single-crystal X-ray diffraction (SCXRD) measurements and crystal structure analysis.** Before the measurements, crystals were stored in a 2:1 (v:v) mixture of *o*-DCB and ionic liquid (methyltrioctylammonium bis(trifluoromethylsulfonyl)imide, $[N_{8881}][NTf_2]$; purity: 99%; supplier: Iolitec) to prevent desolvation of *o*-DCB crystalline solvent during handling. A suitable crystal was picked up and covered with a droplet of *o*-DCB and ionic liquid mixture to mount on a specimen pin using inert oil (LV CryoOil, MiTeGen) for SCXRD measurements. SCXRD measurements of **TK-COF-4** and **-5** were conducted using a single-crystal X-ray diffractometer (XtaLAB Synergy-DW, Rigaku) controlled by CrysAlisPro® software with X-ray focusing mirrors using Cu Kα radiation ($\lambda = 1.54184$ Å) at 93.15 K. Crystal structures of **TK-COF-4** and **-5** were solved by SHELXT, and the least-squares refinement was carried out using SHELXL and Olex2® softwares using restraints of DFIX, SADI, ISOR, and FLAT. Non-hydrogen atoms were refined with an anisotropic thermal factor, and hydrogen atoms were located at calculated positions that were treated using an isotropic riding-atom mode. The disorder of *o*-DCB solvent molecules was treated by the solvent mask method using Olex2® software. The crystallographic tables are presented in Supplementary Sections 2.7 and 2.8 (Supplementary Tables 3 and 4). The face indexes of the crystals of **TK-COF-4** and **-5** were determined by CrysAlisPro® software using the photographs of the specimen crystals and their crystal orientations. See Supplementary Section 2.9 for the results.

**High-resolution transmission electron microscopy (HR-TEM).** We used a spherical-aberration-corrected transmission electron microscope (R005, JEOL) operated at an acceleration voltage of 80 kV. To reduce the sample thickness to ensure electron beam transmission, the sample crystals were mechanically treated according to the method described above. After this treatment, the solvent was replaced with toluene. The toluene-suspended samples were drop-casted onto an ultrathin carbon-coated TEM grid (3150 C, ALLIANCE Biosystems; carbon thickness: 5 nm, grid material: copper, mesh density: 300), which was dried in a vacuum at 80 °C for 12 h with a flow of 50 sccm of dry nitrogen before observation.

**Thermogravimetric analysis (TGA).** We used a thermogravimetric differential thermal analyzer (Thermo Plus EVO2, Rigaku) with a flow of 150 mL min$^{-1}$ of air or nitrogen at a heating rate of 5 °C min$^{-1}$. Samples were dried in a vacuum at 80 °C for 12 h with a flow of 50 sccm of dry nitrogen before measurements. Typically, 3–5 mg of sample was loaded in a platinum pan.

**Gas adsorption isotherm measurements and analyses.** Adsorption-desorption isotherms of the samples were acquired using 3Flex (Micromeritics). The sample crystals, dried following the method described above, were further degassed before measurements according to the following procedures. First, the sample glass tube, in which the sample (typically 80–100 mg) was loaded, was evacuated under vacuum at 90 °C for 6 h using our home-built vacuum equipment. Then, the glass tube was mounted in 3Flex and further degassed

under ultrahigh vacuum using an equipped mantle heater at 100 °C for 12 h. Nitrogen adsorption-desorption isotherms were acquired at 77, 273, and 298 K using high-purity $N_2$ gas (purity: >99.9995%). $CO_2$ adsorption-desorption isotherms were acquired at 273, 285.5, and 298 K using high-purity $CO_2$ gas (purity: 99.999%). Between successive measurements conducted under different conditions, we again degassed the sample using the equipped mantle heater at 100 °C for 2–4 h.

The surface areas were estimated by fitting the $N_2$ adsorption isotherms at 77 K (*cf.* Figure 3c) with the Brunauer–Emmett–Teller (BET) theory using the SESAMI 1 algorithm[59] in the SESAMI® web interface[60] satisfying the Rouquerol criteria. The theoretical pore size distributions were calculated from the crystal structure data using Zeo++® software[61] version 0.2.0 with a 1.67-Å radius probe $N_2$ molecule, the high-accuracy flag, and 100,000 Monte Carlo samples per unit cell. The pore size distributions were also derived using 3Flex's software version 5.01 and applying non-local density functional theory (NLDFT) to the measured $N_2$ adsorption isotherms with the model of "N2–Tarazona NLDFT (Esf: 30.0 K)" and geometry of "Cylinder."

**Calculations of $CO_2/N_2$ adsorption selectivity.** First, the adsorption isotherms of $CO_2$ and $N_2$ at 273 and 298 K acquired for **TK-COF-4** and **-5** (see Supplementary Figs. 24 and 25) were fitted using the following Freundlich–Langmuir equation[62]

$$n = \frac{a \cdot b \cdot p^c}{1 + b \cdot p^c} \quad (1)$$

where $n$ [mmol g$^{-1}$] is the amount adsorbed (the loading), $a$ [mmol g$^{-1}$] is the maximal loading, $p$ [kPa] is the pressure, and $c$ (dimensionless) and $b$ [kPa$^{-c}$] are constants. The values of $n$ were determined by fitting this equation to the acquired adsorption isotherms (see Supplementary Fig. 26 and Table 8).

The $CO_2/N_2$ adsorption selectivity was calculated using the ideal adsorbed solution theory (IAST)[44,45] represented by

$$S = \frac{n_{CO_2}/n_{N_2}}{m_{CO_2}/m_{N_2}} \quad (2)$$

where $n_{CO_2}$ and $n_{N_2}$ [mmol g$^{-1}$] are the molar loading in the adsorbed phase, and $m_{CO_2}$ and $m_{N_2}$ [kPa] are the partial pressures in the bulk gas phase of $CO_2$ and $N_2$, respectively. In the present case, $m_{CO_2}/m_{N_2} = 15/85 \cong 0.1765$. $n_{CO_2}$ and $n_{N_2}$ were calculated using the Freundlich–Langmuir equation above.

**Evaluations of heat of adsorption ($Q_{st}$).** The heat of absorption at the adsorption amount $n$, denoted $Q_{st}(n)$, was determined based on the Clausius-Clapeyron equation

$$Q_{st}(n) = -RT^2 \left( \frac{\partial \ln p}{\partial T} \right)_n \quad (3)$$

where $T$ [K] is the temperature, $p$ is the pressure [Pa], and $R$ is the gas constant. Specifically, we used $CO_2$ adsorption isotherms acquired at three different temperatures (273, 285.5, and 298 K; Supplementary Fig. 24) and 3Flex's software (version 5.01) to calculate $Q_{st}(n)$ for each $n$, which we displayed in the unit of cm$^3$ g$^{-1}$ STP in Fig. 3f.

## Data availability

Crystallographic data for the structures of COFs reported in this article have been deposited at the Cambridge Crystallographic Data Centre (CCDC) under the deposition numbers of 2361003 (**TK-COF-4**), 2361014 (**TK-COF-5**), and 2383526 (**TK-COF-5_dried**). Copies of the data can be obtained free of charge from https://www.ccdc.cam.ac.uk/structures/. The structural data are also presented in Supplementary

Tables 3, 4, and 7. Additional data are available from the corresponding author upon request.

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

## Acknowledgements

This work was supported by JSPS KAKENHI Grants JP22K18286 and JP23H00165 (Y.M.), JP22K05032 and JP24H00005 (H.U.), and JP22H05033 (T.S.). This work was supported in part by the Cooperative Research Program of "Network Joint Research Center for Materials and Devices" from MEXT. We thank the support from the Tokyo Tech GXI (Science Tokyo GXI after October 2024) program from MEXT. We also thank the Open Facility Center at Tokyo Institute of Technology (Institute of Science Tokyo after October 2024) for support provided for the measurements and Ms. Yuko Kishida at Tokyo Institute of Technology for the support provided for determining the crystal face indexes. We cordially appreciate Dr. Alicia Glatfelter for her improving the English sentences of our manuscript.

## Author contributions

Y.M. conceived this research, led this project, predicted unreacted primary amines from the layered structure, and wrote this manuscript. T.Kitano discovered the structure of the present COFs by SCXRD, conducted most

of the experiments and measurements, and wrote part of this manuscript. S.G. first synthesized **TK-COF-5** and contributed to the early stage of this project. X.W. contributed to gas adsorption isotherm, TGA, PXRD measurements, and construction of structural models using Materials Studio® software. H.U. carried out the crystallographic analyses and interpretations related to the structure determinations of the COFs. T.S. and Y.K. conducted HR-TEM observations and pertinent data analyses. T.Kamihara conducted ss-NMR measurements and pertinent data interpretations. Y.S. conducted XPS measurements and pertinent data analysis.

## Competing interests

The authors declare no competing interests.
