## [Peer Review File · Nature Communications]

REVIEWER COMMENTS

Reviewer #1 (Remarks to the Author):

The authors have carefully addressed the reviewers' questions and concerns. The quality of this work has been improved. Publication of this work on Nat. Commun. is recommended.

Reviewer #3 (Remarks to the Author):

[Note from the Editor: Reviewer #3 was asked to look also over the response given to Reviewer #2]

This study presents TK-COF-4 and -5, a new class of 2.5D COFs synthesized using a combination of tetratopic and tritopic building blocks. Unlike previously reported COFs constructed with the same type of building blocks, which resulted in 3D structures, these new COFs exhibit a layered structure due to their unique in-plane interpenetration, where two laterally displaced 2.5D skeletons interlock. Another notable structural characteristic is the presence of high-density, unreacted primary amines pointing perpendicular to the network plane. This out-of-plane arrangement allows for potential modifications with larger chemical moieties. These COFs were found to form large single crystals, exceeding the sizes previously reported for 2D COFs. TK-COF-4 and -5 exhibited high thermal stability, microporosity, notable CO₂ uptake capacity, and good selectivity over N₂. However, their CO₂ adsorption enthalpy was relatively low, which is a common characteristic of COFs with aromatic amine groups. While the 2.5D structure presents a novel approach to COF design, further investigations are needed to explore the generalizability of this method and the impact of amine oxidation on the COFs' long-term performance in CO₂ capture applications. The revisions significantly improved the clarity and accuracy of the manuscript. The inclusion of additional experiments, data analysis, and comparisons with existing literature strengthened the authors' claims and highlighted the potential impact of their 2.5D COF. The manuscript may be adjudged for publication after considering the following comments:

- The high area density of primary amines, surmised to contribute to effective CO₂ capture, lacks a full quantitative comparison to other COFs with similar functionalities. A more comprehensive discussion of how this COF's selectivity and adsorption efficiency compare to known standards in the field is necessary for claims of better performance. Also a discussion on the binding mechanism will make it more scientifically persuasive.
- Expand the discussion to clarify the limitations of 2D and 3D COFs that the 2.5D structure purportedly resolves. This could involve direct comparisons with recent single-crystal COF structures (e.g., Deng et al., 2024, and Wang et al., 2018) and why their methods or structural solutions may not be directly transferable to larger-scale or 2D-like COFs.
- The manuscript describes the synthesis of two COFs using two tritopic building blocks (TFPT and

TFPB) combined with a tetrahedral linker. However, the feasibility of expanding this synthesis approach to other building block combinations has not been fully explored. If testing additional building blocks is not practical, the authors should at least include a discussion on what characteristics of the chosen building blocks (e.g., bonding angles, steric factors) likely contributed to the formation of the 2.5D structure.

Point-by-point responses to the reviewer's comments

Reviewer #3:

This study presents TK-COF-4 and -5, a new class of 2.5D COFs synthesized using a combination of tetratopic and tritopic building blocks. Unlike previously reported COFs constructed with the same type of building blocks, which resulted in 3D structures, these new COFs exhibit a layered structure due to their unique in-plane interpenetration, where two laterally displaced 2.5D skeletons interlock. Another notable structural characteristic is the presence of high-density, unreacted primary amines pointing perpendicular to the network plane. This out-of-plane arrangement allows for potential modifications with larger chemical moieties. These COFs were found to form large single crystals, exceeding the sizes previously reported for 2D COFs. TK-COF-4 and -5 exhibited high thermal stability, microporosity, notable CO₂ uptake capacity, and good selectivity over N₂. However, their CO₂ adsorption enthalpy was relatively low, which is a common characteristic of COFs with aromatic amine groups. While the 2.5D structure presents a novel approach to COF design, further investigations are needed to explore the generalizability of this method and the impact of amine oxidation on the COFs' long-term performance in CO₂ capture applications. The revisions significantly improved the clarity and accuracy of the manuscript. The inclusion of additional experiments, data analysis, and comparisons with existing literature strengthened the authors' claims and highlighted the potential impact of their 2.5D COF. The manuscript may be adjudged for publication after considering the following comments:

Authors' response:

Thank you very much for reviewing our manuscript carefully and giving us insightful comments. Below, we have responded to your comments. Please note that all modifications in this revision are shown in **red**. Sentences in **green** are those from our initially submitted manuscript. We use the abbreviation "SI" to denote "Supplementary Information."

Reviewer 3's comment #1-1:

The high area density of primary amines, surmised to contribute to effective CO₂ capture, lacks a full quantitative comparison to other COFs with similar functionalities.

Authors' response:

Thank you for the comment. As pointed out, our manuscript lacked the quantitative comparison of the area density of the primary amines of our **TK-COF-4/-5** to those of other COFs that had primary amines on their frameworks. To address this comment, we conducted a literature survey, from which we found that COFs (or 2D-COFs) that had primary amines on the framework can be divided into two groups. The first group is 2D-COFs with primary amines left as unreacted moieties (like **TK-COF-4/-5** in this study), which are **PT-/PY-COFs** by Lotsch et al., *Nature Commun.* **10**, 2689 (2019) (DOI: 10.1038/s41467-019-10574-6; ref. 18 in our manuscript) and **NH₂-Th-Tz COF** by Cai et al., *ACS Appl. Mater. Interfaces* **15**, 24836 (2023) (DOI: 10.1021/acsami.3c02025; ref. 20 in our manuscript). The second group is 2D-COFs to which primary amines were appended by post-synthesis modifications, which are **COF-609** by Yaghi et al., *J. Am. Chem. Soc.* **144**, 12989 (2022) (DOI: 10.1021/jacs.2c05382); **Me₃TfB-(NH₂)₂BD** by Smet et al., *ACS Appl. Mater. Interfaces*, **15**, 5118 (2023) (DOI: 10.1021/acsami.2c17672); **COF-284/-285-NH₂** by Yaghi et al., *J. Am. Chem. Soc.* **146**, 89 (2024) (DOI: 10.1021/jacs.3c11688); and **COF-999** by Yaghi et al., *Nature* (2024) (DOI: <https://doi.org/10.1038/s41586-024-08080-x>).

To address the above comment, we have created a new Section S3.10 with a new Table S7 in the revised SI, in which we compared the area densities of primary amines in these COFs, as shown below.

S3.10 Area densities of primary amines of TK-COF-4/-5 and other COFs

We calculated the area densities of primary amines in **TK-COF-4** and **-5** per layer from their structures determined from the SCXRD data analyses. We also calculated the area densities of primary amines in previous COFs per layer from their reported CIF or atomic coordinates. We compared these area densities in Table S7. See the Results and Discussion section in the main text for discussion.

Supplementary Table S7. Comparison of the area density of primary amines in **TK-COF-4/-5** with those in other COFs

COF name	Type	Area density of primary amines [$\times 10^5 \mu\text{m}^{-2}$]	Ref.
TK-COF-4	Unreacted primary amines remaining after the synthesis of COFs	4.9	This work
TK-COF-5		4.6	
PT-/PY-COF		1.6 [†]	S15
NH₂-Th-Tz COF		1.7 [†]	S16

COF-609	Primary amines appended to 2D-COFs by post-synthetic modification	5.3 [†]	S17
Me₃TFB-(NH₂)₂BD		8.0 [†]	S18
COF-284-NH₂		15 [†]	S19
COF-285-NH₂		9.2 [†]	
COF-999		5.1 [†]	S20

[†] Calculated by us from the reported COF structure using Materials Studio[®] software.

To conform with this modification, we have added the following remark to the revised manuscript.

<Page 6 in the revised manuscript>

Before modification:

For the latter, the high area density of the amines (4.9 and $4.6 \times 10^5 \mu\text{m}^{-2} \text{layer}^{-1}$ for **TK-COF-4** and **-5**, respectively) is noted.

After modification:

For the latter, the high area density of the amines (4.9 and $4.6 \times 10^5 \mu\text{m}^{-2} \text{layer}^{-1}$ for **TK-COF-4** and **-5**, respectively) is noted. This area density is much higher compared to previous 2D-COFs in which primary amines were left unreacted, whereas it is comparable to or lower than that of 2D-COFs to which primary amines were appended by post-synthesis modification (Table S7).

To conform to this change, we have added the following reference in the revised SI.

S15. Banerjee, T., Haase, F., Trenker, S., Biswal, B. P., Savasci, G., Duppel, V., Moudrakovski, I., Ochsenfeld, C. & Lotsch, B. V. Sub-stoichiometric 2D covalent organic frameworks from tri- and tetratopic linkers. *Nature Commun.* **10**, 2689 (2019).

S16. Tang, X., Yang, Y., Li, X., Wang, X., Guo, D., Zhang, S., Zhang, K., Wu, J., Zheng, J., Zheng, S., Fan, J., Zhang, W. & Cai, S. Postmodification of an amine-functionalized covalent organic framework for enantioselective adsorption of tyrosine. *ACS Appl. Mater. Interfaces* **15**, 24836–24845 (2023).

S17. Lyu, H., Li, H., Hanikel, H., Wang, K. & Yaghi, O. M. Covalent organic frameworks for carbon dioxide capture from air. *J. Am. Chem. Soc.* **144**, 12989–12995 (2022).

S18. Dautzenberg, E., Li, G. & de Smet, L. C. P. M. Aromatic amine-functionalized covalent organic frameworks (COFs) for CO₂/N₂ separation. *ACS Appl. Mater. Interfaces* **15**, 5118–5127 (2023).

S19. Han, X., Zhou, Z., Wang, K., Zheng, Z., Neumann, S. E., Zhang, H., Ma, T. & Yaghi, O. M. Crystalline

polyphenylene covalent organic frameworks. *J. Am. Chem. Soc.* **146**, 89–94 (2024).

S20. Zhou, Z., Ma, T., Zhang, H., Chheda, S., Li, H., Wang, K., Ehrling, S., Giovine, R., Li, C., Alawadhi, A. H., Abduljawad, M. M., Alawad, M. O., Gagliardi, L., Sauer, J. & Yaghi, O. M. Carbon dioxide capture from open air using covalent organic frameworks. *Nature* (2024). (DOI: <https://doi.org/10.1038/s41586-024-08080-x>)

Reviewer 3's comment #1-2:

A more comprehensive discussion of how this COF's selectivity and adsorption efficiency compare to known standards in the field is necessary for claims of better performance.

Authors' response:

Thank you for the comments. Although we have already compared the CO₂/N₂ selectivity vs. the lowness of the heat of adsorption (*i.e.*, energy efficiency) in Fig. 3g for COFs and in Fig. S36 for MOFs, to address the comment above, we have extended the range of our comparison within the field of organic porous materials. Specifically, we have newly compared our results with those reported for porous organic polymers (POPs), a category of materials competing with COFs, to make our discussion more comprehensive. To do this, we have created a new Section 4.3 in the revised SI as follows, in which we have included a new Fig. S37.

S4.3 POPs

The CO₂/N₂ separation performance of **TK-COF-4**, which exhibited better performance than **TK-COF-5** in this report, is compared with those reported for porous organic polymers (POPs). For the data of **TK-COF-4**, we used the IAST CO₂/N₂ selectivity at a 15:85 ratio ($S_{CN(15:85)}$) and $Q_{st(av)}$, shown in Figs. 3e and 3f in the main text, respectively. For the data of POPs, we used the data shown in Table 1 of Ref. S37, for which the values of $S_{CN(15:85)}$ and Q_{st} were reported. The result is shown in Fig. S37 below. Because the pressures at which $S_{CN(15:85)}$ were evaluated for the POP reports were often not clearly indicated in the literature, we have plotted the values of $S_{CN(15:85)}$ at both 0 and 1 bar for **TK-COF-4** in this figure. According to this comparison, the present **TK-COF-4** has better performance generally than those POPs in the sense that the former realized rather low Q_{st} while retaining sufficiently high $S_{CN(15:85)}$ for CO₂ separation from most industrial flue gases.

Supplementary Figure S37. Comparison of the values of IAST CO_2/N_2 selectivity at a 15:85 ratio and Q_{st} reported for previous POPs presented in Table 1 of Ref. S37 and those for **TK-COF-4** presented in Figs. 3e and 3f in the main text, where $Q_{st} = Q_{st(av)}$.

To conform with this addition, we have made the following modification in the revised manuscript.

<Page 8 in the revised manuscript>

Before modification:

The advantage is also seen when compared with previously reported MOFs (Supplementary Section 4.2).

After modification:

The advantage is also seen when compared with previously reported MOFs (Supplementary Section 4.2) and porous organic polymers (Supplementary Section 4.3).

To conform to this change, we have added the following reference in the revised SI.

S37. Song, K. S., Fritz, P. W. & Coskun, A. Porous organic polymers for CO₂ capture, separation and conversion. *Chem. Soc. Rev.* **51**, 9831–9852 (2022).

Reviewer 3's comment #1-3:

Also a discussion on the binding mechanism will make it more scientifically persuasive.

Authors' response:

We consider that the present binding mechanism is physisorption, as judged from the low Q_{st} ($\approx 25 \text{ kJ mol}^{-1}$) and the Langmuir-type adsorption isotherms. Previously, the mechanism of CO₂ capture by primary amines attached to aromatic rings of porous organic polymers was concluded to be physisorption based on the low Q_{st} ($\leq 30 \text{ kJ mol}^{-1}$) and the Langmuir-type adsorption isotherms (El-Kaderi et al, *J. Mater. Chem. A* **1**, 10259 (2013); de Smet et al, *ACS Appl. Mater. Interfaces* **15**, 5118 (2023)).

To convey this viewpoint to readers, we have made the following modification in the revised manuscript.

<Page 8 in the revised manuscript>

Before modification:

Such low $Q_{st(av)}$ values are considered to be caused by the low basicity of aromatic amines.

After modification:

Such low $Q_{st(av)}$ values are considered to be caused by the low basicity of aromatic amines, from which we attribute the present capture mechanism to physisorption^{53,54}.

To conform to this change, we have added the following reference in the revised manuscript.

53. İslamoğlu, T., Rabbani, M. G. & El-Kaderi, H. M. Impact of post-synthesis modification of nanoporous organic frameworks on small gas uptake and selective CO₂ capture. *J. Mater. Chem. A* **1**, 10259–10266 (2013).

54. Dautzenberg, E., Li, G. & de Smet, L. C. P. M. Aromatic amine-functionalized covalent organic frameworks (COFs) for CO₂/N₂ separation. *ACS Appl. Mater. Interfaces* **15**, 5118–5127 (2023).

Reviewer 3's comment #2:

Expand the discussion to clarify the limitations of 2D and 3D COFs that the 2.5D structure purportedly resolves. This could involve direct comparisons with recent single-crystal COF structures (e.g., Deng et al., 2024, and Wang et al., 2018) and why their methods or structural solutions may not be directly transferable to larger-

scale or 2D-like COFs.

Authors' response:

Thank you for the constructive comment. This point has been stated in the manuscript from multiple perspectives, such as in the following statements: “So far, the record single-crystal size of 2D-COFs attained with one-step growth has been limited to *ca.* 3–10 $\mu\text{m}^{29,30}$; to attain further larger sizes, a recent X-ray structural study used two-step growth in which seed crystals were re-grown in a fresh solution, yielding pyrene 2D-COF crystals with sizes of 15–45 μm^{13} ,” where ref. 13 is Deng et al., 2024; “Specifically, this structural concept provides a new avenue to simultaneously realize advantages of 2D and 3D COFs, typical of which are a multitude of applications and ease of growing large single crystals, respectively, by reconciling the previously established binary distinction between 2D and 3D concepts;” “The virtue of the 2.5D concept lies in the ability to generate large single crystals owing to the essentially 3D nature (remember Haase and Lotsch’s remark³⁵ above) and hence the effectiveness to address the crystallinity problem that has particularly afflicted 2D-COFs;” and “Among them, (iii) is significant because purposefully chosen chemical moieties could be appended perpendicularly to the covalently reticulated crystalline layers, different from the aforementioned 2D-COFs with unreacted groups in which the size of the appending moieties is, in principle, limited by the pore size of the 2D-COFs.” However, *when considering the Reviewer’s comment above, we noticed that all of these statements are made in the Introduction but not made in the Results & Discussion nor Conclusion sections at all.* We think that this fault, which had been unnoticed by us, may have given the Reviewer an impression of the lack of our discussion on the limitations of 2D- and 3D-COFs that the 2.5D structure resolves.

In essence, we can summarize the “limitations of 2D and 3D COFs that the 2.5D structure resolves” by classifying the arguments into the following three essential aspects:

Aspect 1: 2.5D-COFs can resolve the crystallization problem that has particularly afflicted 2D-COF, for which the largest single-crystal size is still less than 45 μm even when resorting to a time-consuming two-step seed crystallization method (Deng et al., JACS 2024). The underlying reason for the ability to achieve large single crystals is that 2.5D-COFs satisfy the Haase and Lotsch’s criterion—*covalent connectivity in three independent directions can more easily improve crystallinity than 2D-COFs that use two independent directions (Chem. Soc. Rev. 2020).*

Aspect 2: 3D-COFs can achieve large crystal sizes (*e.g.*, Wang et al., Science 2018) but cannot answer the need for a *planar shape* that is useful for broad applications. 2.5D-COFs can provide planar shapes with high crystallinity and large crystal sizes comparable to those of 3D-COFs.

Aspect 3: When unreacted primary amines left on 2D-COFs are to be used as the functionality to which chemical moieties are appended, the size of the moiety is, in principle, limited to the size of the pore. 2.5D-

COFs essentially do not suffer from this limitation because the unreacted primary amines point in out-of-plane directions.

To state explicitly the essential aspects regarding the limitations of 2D and 3D-COFs that the 2.5D structure resolves, we have added the relevant statements to the Conclusion section as follows.

<Page 8–9, Conclusion>

After modification:

This article reported a new class of COFs with 2.5D—microscopically 3D but macroscopically 2D—skeletons, embodied by **TK-COF-4** and **-5**. These COFs have unique features owing to their constitution of tetrahedral–tetratopic building blocks and their non-stoichiometric bonding with trigonal–tritopic building blocks, whereas all previous reports that combined such building blocks have generated 3D-COFs. **Although we posed the hypothesis above,** the range of the generality of this approach or requisites for constructing such 2.5D structures should be clarified in the future as an open question. Because of the structural uniqueness, unreacted free primary amines are, for the first time, aligned perpendicular to the laterally extended networks reticulated by covalent bonds. **This feature has resolved the limitation of 2D-COFs in which the size of the chemical moieties appended to the unreacted functionality is, in principle, limited by the size of the pore.** Owing to the 3D bonds comprising this class of COFs, the record-large single-crystal sizes of up to 0.1 mm were achieved for layered COFs, which is expected to be advantageous for broad applications. **Thus, 2.5D COFs have resolved the limitations of 3D-COFs (a difficulty in providing a planar shape that is useful for applications) and 2D-COFs (a difficulty in achieving large single crystals).** High thermal stability and CO₂ capture performances, as well as the high-density primary amines arranged normally on the planar skeletons, may render these COFs useful for broad applications. Such a new class of COFs will open a new domain of covalently-reticulated crystalline systems, further extending the growing frontier of reticular chemistry.

Reviewer 3's comment #3:

The manuscript describes the synthesis of two COFs using two tritopic building blocks (TFPT and TFPB) combined with a tetrahedral linker. However, the feasibility of expanding this synthesis approach to other building block combinations has not been fully explored. If testing additional building blocks is not practical, the authors should at least include a discussion on what characteristics of the chosen building blocks (e.g., bonding angles, steric factors) likely contributed to the formation of the 2.5D structure.

Authors' response:

Previously, all COFs made by combining tetrahedral and triangular building blocks resulted in 3D-COFs with either **bor** or **ctn** topology, and their reported densities were rather low (0.13–0.53 g cm⁻³). In contrast, the densities of our **TK-COF-4** and **-5** were much higher (*ca.* 0.7–1 g cm⁻³), which implies higher thermal stability of **TK-COF-4/-5**. To address the comment above, we carried out structural energy calculations with the COMPASS III force field using Materials Studio[®] software for **TK-COF-4/-5** and hypothetical 3D-COFs with **bor** and **ctn** topology constructed by connecting **TAM** and **TFPT/TFPB**, and we compared these calculated energies.

As a result, we found that the total energies of **TK-COF-4/-5** were much lower than those of these hypothetical 3D-COFs *because the non-bonding (such as van der Waals and electrostatic) energies of TK-COF-4/-5 were much lower than those of the hypothetical bor and ctn 3D-COFs*. Based on these calculational results, we proposed a hypothesis that the possibility of forming 2D-COFs would be higher if one uses building-block molecules that give rise to large *inter-layer* van der Waals attractions (such as $\pi\cdots\pi$) and hydrogen bonds (such as N–H \cdots N) *when the 2.5D structure is formed*. To include this discussion and hypothesis, we have added the new paragraph below just before the Conclusion section in the revised manuscript.

<Page 8 of the revised manuscript>

We have added the following paragraph.

Finally, we comment on the factors that may contribute to the formation of the 2.5D structure. Previous COFs made by combining tetrahedral and triangular building blocks were 3D-COFs with either **bor**^{5,40} or **ctn**^{5,39} topology, the reported densities of which were low (0.17–0.41 g cm⁻³ in ref. 5; 0.43–0.53 g cm⁻³ in ref. 39; 0.13 g cm⁻³ in ref. 40). In contrast, the density of **TK-COF-4/-5** was much higher (*ca.* 0.7–1 g cm⁻³, Table S6), implying higher thermodynamic stability. From our structural energy calculations using the COMPASS III force field, the total energies of **TK-COF-4/-5** were much lower than those of the hypothetical **bor** and **ctn** COFs constructed using **TAM** and **TFPT/TFPB** *because the non-bonding energies of TK-COF-4/-5 were much lower than those of these hypothetical 3D-COFs* (Supplementary Section 3.15). Therefore, we hypothesize that the use of building blocks that would cause large *inter-layer* van der Waals attractions (such as $\pi\cdots\pi$) and hydrogen bonds (such as N–H \cdots N) shall increase the possibility of forming the 2.5D structure.

To conform with this addition, we have created two new Sections in the revised SI; they are Sections S3.9, in which a new Table S6 has been added, and S3.15, in which new Tables S10 and S11 have been added, as follows.

S3.9 Framework densities of TK-COF-4/-5 and hypothetical 3D-COFs with bor and ctn topology

We calculated the densities of TK-COF-4/-5 and those of hypothetical COFs that have **bor** and **ctn** topology constructed by connecting TFPT/TFPB and TAM using Materials Studio[®] software. We used the COMPASS III force field for their geometrical optimizations. We also calculated the densities of TK-COF-4/-5 from the structures determined by SCXRD measurements. These results are compared in Table S6. The densities of TK-COF-4 and -5 (*ca.* 0.7–1 g cm⁻³) are much higher than those of the hypothetical COFs (< 0.15 g cm⁻³). Note that the previously reported 3D-COFs with **bor** and **ctn** topology actually had low densities (see the final paragraph of the Results and Discussion section in the main text).

Table S6. Comparison of the density of TK-COF-4/-5 with those of hypothetical COFs with **bor** and **ctn** topology constructed by connecting TFPT/TFPB and TAM

COF type, Space group	Density calculated from the structure determined by SCXRD [g cm ⁻³]	Density of the model after geometrical optimization by Materials Studio [®] [g cm ⁻³]
TK-COF-4 , P2₁/n (No. 14)	0.741 [†]	0.685
Hypothetical bor -topological COF made from TFPT and TAM, P23 (No. 195)	—	0.124
Hypothetical ctn -topological COF made from TFPT and TAM, I-43d (No. 220)	—	0.137
TK-COF-5 , P2₁/n (No. 14)	0.684 [†]	0.744
TK-COF-5_dried , P2₁/n (No. 14)	1.03	1.09
Hypothetical bor -topological COF made from TFPB and TAM, P23 (No. 195)	—	0.119
Hypothetical ctn -topological COF made from TFPB and TAM, I-43d (No. 220)	—	0.130

[†] Solvent molecules were excluded in the density calculations.

S3.15 Energy calculations of the frameworks

We calculated the energies of **TK-COF-4/-5** and the hypothetical COFs with **bor** and **ctn** topology constructed by connecting **TFPT/TFPB** and **TAM**. The geometrical optimizations were conducted using COMPASS III force field with Materials Studio[®] software. Because the chemical formula and molar mass of these COFs are different, we calculate and show the energies in unit of kJ kg^{-1} so that the energies for different COFs are intercomparable. The chemical formula and molar mass of those COFs are as follows.

- **TK-COF-4**: chemical formula = $\text{C}_{196}\text{H}_{132}\text{N}_{28}$, molar mass = $2879.41 \text{ g mol}^{-1}$.
- **TK-COF-5/-5_dried**: chemical formula = $\text{C}_{208}\text{H}_{144}\text{N}_{16}$, molar mass = $2867.55 \text{ g mol}^{-1}$.
- Hypothetical COF with **bor** topology constructed from **TFPT** and **TAM**: chemical formula = $\text{C}_{171}\text{H}_{108}\text{N}_{24}$, molar mass = $2498.91 \text{ g mol}^{-1}$.
- Hypothetical COF with **ctn** topology constructed from **TFPT** and **TAM**: chemical formula = $\text{C}_{684}\text{H}_{432}\text{N}_{96}$, molar mass = $9995.65 \text{ g mol}^{-1}$.
- Hypothetical COF with **bor** topology constructed from **TFPB** and **TAM**: chemical formula = $\text{C}_{183}\text{H}_{120}\text{N}_{12}$, molar mass = $2487.06 \text{ g mol}^{-1}$.
- Hypothetical COF with **ctn** topology constructed from **TFPB** and **TAM**: chemical formula = $\text{C}_{732}\text{H}_{480}\text{N}_{48}$, molar mass = $9948.23 \text{ g mol}^{-1}$.

As summarized in Tables S10 and S11, the results revealed that the energies—especially non-bond energies—of **TK-COF-4** and **-5** are much lower than those of the hypothetical 3D-COF with **bor** and **ctn** topology, implying that **TK-COF-4** and **-5** are thermodynamically more stable than these hypothetical 3D-COFs.

Table S10. Energies calculated for **TK-COF-4** and the corresponding hypothetical **bor-** and **ctn-** topological COFs constructed by connecting **TFPT** and **TAM**

	This work	Hypothetical COFs	
	TK-COF-4 [kJ kg ⁻¹]	bor topology [kJ kg ⁻¹]	ctn topology [kJ kg ⁻¹]
Valence energy (diag. terms): {A}	651.53	772.41	785.53
Bond	68.14	85.79	71.69
Angle	883.58	122.70	227.49
Torsion	-307.30	563.04	485.35
Inversion	7.11	0.87	1.01
Valence energy (cross terms): {B}	-80.59	-4.68	-2.64
Stretch-Stretch	4.80	3.36	0.83
Stretch-Bend-Stretch	-9.94	-14.46	-22.90
Stretch-Torsion-Stretch	-24.20	-37.97	-21.78
Separated-Stretch-Stretch	3.69	4.46	1.67
Torsion-Stretch	-114.07	-158.42	-73.63
Bend-Bend	-0.67	0.00	0.00
Torsion-Bend-Bend	-0.90	0.43	-5.75
Bend-Torsion-Bend	60.69	157.92	118.92
Non-bond energy: {C}	-532.72	-16.27	65.63
van der Waals	75.10	254.78	329.64
Long Range Correction	-3.40	-0.58	-0.68
Electrostatic	-604.42	-270.47	-263.33
Total energy = {A} + {B} + {C}	38.22	711.47	848.52

Table S11. Energies calculated for **TK-COF-5** and the corresponding hypothetical **bor-** and **ctn-** topological COFs constructed by connecting **TFPB** and **TAM**

	This work		Hypothetical COFs	
	TK-COF-5 [kJ kg ⁻¹]	TK-COF-5_dried [kJ kg ⁻¹]	bor topology [kJ kg ⁻¹]	ctn topology [kJ kg ⁻¹]
Valence energy (diag. terms): {A}	1148.14	1881.35	2160.75	2157.33
Bond	71.92	98.44	94.93	83.57
Angle	103.92	95.97	89.01	181.39
Torsion	962.47	1677.33	1974.79	1891.15
Inversion	9.83	9.61	2.02	1.22
Valence energy (cross terms): {B}	-84.41	-118.64	-122.82	-77.08
Stretch-Stretch	3.29	4.01	3.66	1.23
Stretch-Bend-Stretch	-5.90	-5.11	-5.39	-16.80
Stretch-Torsion- Stretch	-26.17	-48.95	-53.97	-35.66
Separated-Stretch- Stretch	0.56	-0.15	-1.47	-3.32
Torsion-Stretch	-111.71	-143.17	-138.87	-51.53
Bend-Bend	-0.14	-0.48	0.00	0.00
Torsion-Bend-Bend	-0.74	0.06	-0.50	-7.14
Bend-Torsion-Bend	56.41	75.15	73.73	36.13
Non-bond energy: {C}	-166.99	-226.86	240.77	317.12
van der Waals	54.08	-28.45	257.25	336.93
Long Range Correction	-3.71	-5.38	-0.58	-0.67
Electrostatic	-217.36	-193.02	-15.90	-19.13
Total energy = {A} + {B} + {C}	896.74	1535.85	2278.70	2397.36

Concomitant to this modification, we made the following expressional modification.

<Page 7, the second paragraph of the revised manuscript>

Before modification:

Finally, to exemplify a potential application,

After modification:

To exemplify a potential application,

Finally, below is the list of minor spontaneous modifications that we have made in this revision.

<Page 2 in the revised manuscript> Correction of the reference number.

Before modification:

... albeit 3D-COF crystals over 100 μm are still rare^{13,30,32,34}.

After modification:

... albeit 3D-COF crystals over 100 μm are still rare^{30,32,34}.

We noticed an erroneous sentence, perhaps caused by an unconscious copy-and-paste of words into a sentence during writing, at the end of the introductory section. We have corrected it as follows.

<Page 3 in the revised manuscript>

Before modification:

... (iii) have high-density unreacted functionalities that point perpendicular to the planar network, and (iv) exhibit *in-plane* interpenetration formed by laterally displaced networks. Among them, (iii) have high-density primary amine groups pointing *out-of-plane*, and (iv) exhibit *in-plane* interpenetration formed by laterally displaced networks. Among them, (iii) is significant because...

After modification:

... (iii) have high-density unreacted functionalities that point perpendicular to the planar network, and (iv) exhibit *in-plane* interpenetration formed by laterally displaced networks. Among them, (iii), ~~have~~ high-density primary amine groups pointing *out-of-plane*, ~~and (iv) exhibit *in-plane* interpenetration formed by laterally displaced networks. Among them, (iii)~~ is significant because...

We noticed miscellaneous expressional mistakes in the graphic of Figure 2. We have corrected this figure as follows.

Before modification:

After modification:

(Explanation of the modifications: In panel (a) of “before modification,” a light-blue frame was missing in the magnified graphic that shows the hydrogen bond in TK-COF-4. In panel (c) of “before modification,” a pink arrow was missing in the magnified TEM image of TK-COF-5.)

We have amended the author contributions descriptions for Y. M., T. Kitano, and X. W. to describe their contributions more accurately and specifically as follows. The authors confirmed and agreed with these modifications.

<Page 13–14, Author contributions section>

Before modification:

Y. M. conceived this research, led this project and wrote this manuscript. T. Kitano discovered the structure of the present COFs, conducted most of the experiments and measurements, ... X. W. contributed to gas adsorption isotherm, TGA, and PXRD measurements.

After modification:

Y. M. conceived this research, led this project, predicted unreacted primary amines from the layered structure, and wrote this manuscript. T. Kitano discovered the structure of the present COFs by SCXRD, conducted most of the experiments and measurements, ... X. W. contributed to gas adsorption isotherm, TGA, PXRD measurements, and construction of structural models using Materials Studio® software.

We heartily appreciate your reviewing our manuscript and providing us with valuable comments. We believe that your insightful, constructive comments have highly improved the quality of our manuscript.

REVIEWERS' COMMENTS

Reviewer #3 (Remarks to the Author):

The revised manuscript comprehensively addresses the comments raised by me, improving its scientific clarity and depth. All pertinent points raised earlier have been analysed and appropriate citations have also been included. Key modifications include quantitative comparisons of primary amine densities (new Section S3.10, Table S7) and extended performance evaluations of TK-COF-4/-5 versus MOFs and porous organic polymers (POPs) (new Section S4.3, Figure S37). The authors clarified the CO₂ capture mechanism as physisorption based on Q_{st} values and isotherm analysis, and they expanded the discussion on 2D and 3D COF limitations, highlighting how 2.5D COFs overcome challenges such as small crystal sizes and functional limitations. Energy calculations and a hypothesis about building block interactions (Sections S3.9 and S3.15) further support their claims.

Additionally, conclusions about 2.5D COFs resolving 2D and 3D COF limitations should be framed as hypotheses rather than definitive outcomes. While comparisons focus on selectivity and adsorption efficiency, practical factors like synthesis scalability, and stability remain underexplored. Overall, except these minor points the revisions have substantially strengthened the manuscript, and the manuscript is well-prepared for publication without further modifications.

Response to Reviewer 3's comments

Reviewer #3:

The revised manuscript comprehensively addresses the comments raised by me, improving its scientific clarity and depth. All pertinent points raised earlier have been analysed and appropriate citations have also been included. Key modifications include quantitative comparisons of primary amine densities (new Section S3.10, Table S7) and extended performance evaluations of TK-COF-4/-5 versus MOFs and porous organic polymers (POPs) (new Section S4.3, Figure S37). The authors clarified the CO₂ capture mechanism as physisorption based on Q_{st} values and isotherm analysis, and they expanded the discussion on 2D and 3D COF limitations, highlighting how 2.5D COFs overcome challenges such as small crystal sizes and functional limitations. Energy calculations and a hypothesis about building block interactions (Sections S3.9 and S3.15) further support their claims.

Additionally, conclusions about 2.5D COFs resolving 2D and 3D COF limitations should be framed as hypotheses rather than definitive outcomes. While comparisons focus on selectivity and adsorption efficiency, practical factors like synthesis scalability, and stability remain underexplored. Overall, except these minor points the revisions have substantially strengthened the manuscript, and the manuscript is well-prepared for publication without further modifications.

Authors' response:

Again, we greatly appreciate your reviewing our manuscript. I, on behalf of the authors, am glad to hear that “the manuscript is well-prepared for publication without further modifications.”

We have addressed your minor suggestion above, “conclusions about 2.5D COFs resolving 2D and 3D COF limitations should be framed as hypotheses rather than definitive outcomes” as follows.

<Page 9, in the summary paragraph>

Before modification:

Thus, 2.5D COFs have resolved the limitations of 3D-COFs (a difficulty in providing a planar shape that is useful for applications) and 2D-COFs (a difficulty in achieving large single crystals).

After modification:

Thus, 2.5D COFs **are considered to** have resolved the limitations of 3D-COFs (a difficulty in providing a planar shape that is useful for applications) and 2D-COFs (a difficulty in achieving large single crystals).

Additionally, because we found some lack of descriptions in Table S1 (previous Table S2), we have amended the lacks in this final revision; this amendment never affects the conclusions of this report. Also, we corrected a few typos, in which we mistakenly wrote 293 K, which is correctly 298 K.

Thank you again for your careful reading of our manuscript and providing insightful comments.